# Stimulus-Responsiveness of Thermo-Sensitive Polymer Hybridized with N-Doped Carbon Quantum Dots and Its Applications in Solvent Recognition and Fe^3+^ Ion Detection

**DOI:** 10.3390/polym14101970

**Published:** 2022-05-12

**Authors:** Tong Chen, Hongwei Zhang, Sanping Zhao

**Affiliations:** State Key Laboratory of New Textile Materials and Advanced Processing Technologies, Wuhan Textile University, Wuhan 430073, China; chentong1997518@163.com (T.C.); hwzhang@wtu.edu.cn (H.Z.)

**Keywords:** N-doped carbon quantum dots, thermo-sensitive polymer, phase transition temperature, solvent recognition, ion detection

## Abstract

To fabricate N-CQDs hybrid thermo-sensitive polymer (poly-N-CQDs), N-doped carbon quantum dots (N-CQDs) with strong blue fluorescence and poly(*N*-isopropylacrylamide-*co*-acrylic acid) (poly(NIPAAm-*co*-AAc)) copolymer with thermo-sensitivity were synthesized, respectively. Subsequently, the coupling reaction between. the -COOH groups of poly(NIPAAm-*co*-AAc) and the -NH_2_ groups on the surface of the N-CQDs was carried out. The fluorescence spectra show that the coil-globule transition of the poly-N-CQDs coincided with intensity changes in the scattering peak at excitation wavelength with the temperature variations. The phase transition temperature and the fluorescent intensity of poly-N-CQDs can be regulated by modulating the composition and concentration of poly-N-CQDs as well as the temperature and pH of the local medium. The thermo-sensitivity and fluorescent properties of the poly-N-CQDs displayed good stability and reversibility. The fluorescence intensity and emission wavelengths of the poly-N-CQDs significantly changed in different solvents for solvent recognition. The poly-N-CQDs was employed as a fluorescent probe for Fe^3+^ detection ranging from 0.025 to 1 mM with a limit of detection (LOD) of 9.49 μM. The hybrid polymer materials have the potential to develop an N-CQDs-based thermo-sensitive device or sensor.

## 1. Introduction

Carbon quantum dots (CQDs), a type of zero-dimensional fluorescent nanomaterials with a size of less than 10 nm, have drawn fantastic attention in the fields of bioimaging [1], fluorescent inks [2], detection of ions and organics [3,4] and biosensors [5] due to their high photostability, superior water solubility, excellent biocompatibility and easily functionalized surface, which can influence excitation, emission and even the quantum yield [6,7]. In recent years, different strategies were developed to improve the fluorescence quantum yield of the CQDs to meet the diversified applications [8,9]. Among them, nitrogen doping or the co-doping of nitrogen with other atoms (boron, phosphorus, sulfur, etc.) are considered to be the most effective approaches to improving fluorescence efficiency [10,11]. Furthermore, nitrogen doping for CQDs can provide rich reactive sites on the surfaces of the CQDs for the chemical modification to broaden the potential applications [12,13].

Luminescent hybrid materials, fabricated via covalent conjugation of CQDs to polymers or physical loading of CQDs into polymer matrices, have enriched the application fields of CQDs, which simultaneously shows optical properties of the CQDs and the stimuli-responsiveness and biocompatibility of polymer matrices [14,15]. Ji et al. [16] prepared a high-strength self-healing poly(vinyl alcohol)-carbon nanodots (PVA-CNDs) hydrogel via physical cross-linking of carbon nanodots and poly(vinyl alcohol); this hydrogel can be used as a detector for Fe^3+^ once the detection limit reached 10 μM. Huang and coworkers [17] introduced carbon nanodots (CDs) based on peach gum polysaccharide into a poly(NIPAAm) hydrogel network to fabricate a luminescent hydrogel. The luminescent hydrogel changed from a highly transparent hydrogel with a strong blue emission under ultraviolet (UV) light to an opaque hydrogel with a weak fluorescent emission when the temperature increased from 20 °C (below lower critical solution temperature (LCST)) to 40 °C (above LCST). Liu et al. [18] synthesized fluorescent CNDs functionalized with glycidyl methacrylate and further fabricated the luminescent hydrogel by copolymerizing with oligo(ethylene glycol) methacrylate and polyethylene glycol diacrylate for sustained release of the drug.

In this work, a novel hybrid polymer with thermo-sensitivity and photoluminescent (PL) property was designed and fabricated. The N-CQDs were synthesized using citric acid (CA) and ethylenediamine (EDA) via a facile one-pot hydrothermal carbonization method, and the poly(*N*-isopropylacrylamide-*co*-acrylic acid) copolymers were prepared via free radical copolymerization, respectively. The N-CQDs hybrid thermo-sensitive polymer (poly-N-CQDs) was obtained via a coupling reaction between the -COOH groups of poly(NIPAAm-*co*-AAc) and the -NH_2_ groups on the surface of the N-CQDs. The thermo-sensitivity and fluorescent properties of the as-prepared poly-N-CQDs was investigated at different compositions and concentrations of poly-N-CQDs as well as temperature and pH of the local medium. The poly-N-CQDs used as a fluorescent probe was investigated for solvent recognition and Fe^3+^ ion detection.

## 2. Experimental

### 2.1. Materials

Citric acid (CA, 99.5%), ethylenediamine (EDA, 98%), acrylic acid (AAc, 99%), N-hydroxysuccinimide (NHS, 98%), 4-(dimethylamino) pyridine (DMAP, 99%) and 1-(3-dimethylaminopropyl)-3-ethylcarbodiimide hydrochloride (EDC, 99%) were purchased from Shanghai Aladdin Biochemical Technology Co., Ltd., China. The *N*-isopropylacrylamide (NIPAAm) (97%, Aldrich, St. Louis, MO, USA) was purified by recrystallization in hexane. The acrylic acid (AAc) (AR, Sinopharm Chemical Reagent Co., Ltd., Shanghai, China) was distilled under a vacuum just before use. The 2,2′-azoisobutyronitrile (AIBN) (AR, Sinopharm Chemical Reagent Co., Ltd., Shanghai, China) underwent recrystallization in methanol three times just before use. The THF (AR, Sinopharm Chemical Reagent Co., Ltd., Shanghai, China) was treated with CaH_2_ for two days and distilled just before use. All other chemicals were of analytical grade and used without further purification.

### 2.2. Synthesis of N-CQDs

The N-CQDs were prepared according to a previously reported hydrothermal method [19]. Briefly, 0.42 g of citric acid (CA) and 0.5 mL of ethylenediamine (EDA) were firstly dissolved in 80 mL of distilled water under stirring. Then, the solution was transferred into a 100-mL Teflon-lined stainless steel autoclave (Shanghai Qiangstrong Industrial Development Co., Ltd., Shanghai, China) and held at 180 °C for 5 h in an oven. After the autoclave naturally cooled to room temperature, the resulting transparent-yellow solution of N-CQDs was purified using a 0.22-μm filter membrane and subjected to dialysis (MWCO = 1000 Da) against distilled water for 12 h. The product was obtained by freeze-drying. Finally, the N-CQDs solution with a concentration of 50 wt% was prepared and stored at 4 °C for future use.

### 2.3. Synthesis of the Thermo-Sensitive Poly(NIPAAm-co-AAc) Copolymers

The poly(NIPAAm-*co*-AAc) copolymers were prepared according to the previously reported method [20]. Two copolymers were synthesized by controlling the feed molar ratio of NIPAAm to AAc ([NIPAAm]/[AAc] is 93:7 and 86:14, respectively). Typically, 12 g of NIPAAm and 1.24 g of AAc, used as monomers with 0.15 g of AIBN as the initiator, were dissolved in 150 mL of dry tetrahydrofuran (THF) in a 250-mL round-bottomed flask. The solution was stirred using a magnetic stirrer (Shanghai Yuezhong Instrument Equipment Co., Ltd., Shanghai, China) under a constant flow of argon for 30 min before polymerization. The reaction was carried out at 75 °C for 3 h. The crude product was precipitated in an excess of anhydrous ethyl ether. To remove the residual unreacted monomers and low-molecular-weight copolymer, the product was dissolved in cold distilled water and precipitated by dropwise addition into distilled water of 60 °C; this process was repeated three times. The final sample was obtained by freeze drying. The as-prepared copolymers are coded as poly_7_ (yield: 89%, GPC: *M*_n_ = 8900, *M*_w_/*M*_n_ = 1.78) and poly_14_ (yield: 87%, GPC: *M*_n_ = 9300, *M*_w_/*M*_n_ = 1.89), respectively.

### 2.4. Preparation of N-CQDs Hybrid Thermo-Sensitive Copolymers (Poly-N-CQDs)

The poly-N-CQDs was synthesized via the coupling reaction between the -COOH groups of poly(NIPAAm-*co*-AAc) copolymer and the -NH_2_ groups on the surface of the N-CQDs, similar to the reported method [21]. Typically, 6 g of the above-mentioned poly_14_ was completely dissolved in 100 mL of distilled water in a 250-mL round-bottomed flask, 2.3 g of EDC was introduced into the solution, and the mixture was reacted for 3 h at room temperature. Then, 1.5 g of NHS was added into the mixture, and the reaction continued for 9 h at room temperature. Finally, 2.5 g of N-CQDs solution (50 wt%) and 1.2 g of DMAP were added into the reaction mixture, and the mixture was reacted at room temperature for another 24 h. The mixture was subjected to dialysis (MWCO = 3500 Da) against distilled water for 7 days. The powder product was obtained by freeze-drying. The powder was dissolved in cold distilled water and precipitated by dropwise addition into distilled water of 60 °C; this process was repeated three times. The final sample was obtained by freeze drying, and the sample was named as poly_14_-N-CQDs (yield: 82%). The poly_7_-N-CQDs (yield: 80%) was prepared according to the same procedure.

### 2.5. Characterization

#### 2.5.1. Fourier Transform Infrared Spectroscopy (FT IR)

FT IR spectra were measured by a Bruker Tensor 27 Fourier transform infrared spectroscopy (Bruker, Ettingen, German) in the 4000–400 cm^−1^ with a resolution of 4 cm^−1^ and a total of 16 scans on KBr pellets for the poly_14_, N-CQDs and poly_14_-N-CQDs samples.

#### 2.5.2. Gel Permeation Chromatography (GPC)

The molecular weight and its polydispersity (*M*_w_/*M*_n_) of the poly_7_ and poly_14_ samples were measured by gel permeation chromatography (GPC) (Agilent 1200, Agilent Technologies, Santa Clara, CA, USA) in THF at room temperature using polystyrene as a standard.

#### 2.5.3. Particle Size Distribution (PSD)

The particle size determinations of dilute aqueous solutions of the poly_14_, poly_7_-N-CQD and poly_14_-N-CQDs samples were analyzed using a Zetasizer Nano ZS ZEN3600 (Malvern Instruments Ltd., Worcestershire, UK) at 32 °C, 40 °C and 50 °C, respectively. Each formulation was analyzed in triplicate.

#### 2.5.4. Transmittance Measurement

The transmittance of the sample solutions was measured by a SHIMADZU UV-2550 spectrophotometer (Shimadzu, Kyoto, Japan) at 500 nm under different temperatures in triplicate.

#### 2.5.5. Transmission Electron Microscopy (TEM)

The morphology of the poly_14_-N-CQDs was obtained using a JEM-2100 transmission electron microscope (JEOL, Tokyo, Japan).

#### 2.5.6. Fluorescence Spectrum

Fluorescence spectra were recorded with an F-2500 fluorescence spectrophotometer (Hitachi, Tokyo, Japan).

The digital photos were taken under a 365-nm UV light.

## 3. Results and Discussion

### 3.1. The Structure and Morphology of N-CQDs, Poly(NIPAAm-co-AAc) and Poly-N-CQDs

The synthetic routes of the N-CQDs, poly(NIPAAm-*co*-AAc) and poly-N-CQDs are shown in Figure 1. Figure 1 exhibits the FT IR spectra of the N-CQDs, poly_14_ and poly_14_-N-CQDs. As seen from Figure 1, the FT IR spectrum of the as-fabricated N-CQDs shows broad absorption bands appearing at 3420 cm^−1^ and 3274 cm^−1^, which could be attributed to the stretching vibrations of O-H and N-H [22], respectively. The two characteristic bands of amide Ι and amide ΙΙ of the amide groups from the N-CQDs are observed at 1657 cm^−^^1^ cm and 1562 cm^−^^1^, respectively. The FT IR spectrum of the poly_14_ displays broad absorption bands at 3300–3500 cm^−1^, which are ascribed to the stretching vibration peaks of N-H from the -NH-CO- groups and O-H from the -COOH groups; the wide peak becomes weaker after reacting with the N-CQDs. The characteristic band at 1730 cm^−1^ is the carbonyl stretching peak of the -COOH groups, which almost disappears in the spectrum of the poly_14_-N-CQDs due to the coupling reaction between the -COOH groups and the -NH_2_ groups. The two characteristic bands of amide Ι and amide ΙΙ of the amide groups from the poly_14_ are observed at 1649 cm^−^^1^ cm and 1553 cm^−^^1^ [23], respectively, and the two peaks become wider in the spectrum of the poly_14_-N-CQDs.

The morphology of the poly_14_-N-CQDs is observed by TEM (as shown in Figure 2 and Appendix A). It can be obviously observed that pristine N-CQDs exhibit an approximately spherical morphology with particle sizes ranging from 4 to 7 nm. Furthermore, it displays a high crystallinity with a lattice distance of approximately 0.32 nm, shown in the insert of Figure 2, which is closed to the interlayer distance between the graphene layers (002 facet) [24]. The above-mentioned results show that N-CQDs are successfully grafted onto the poly(NIPAAm-*co*-AAc) chains via the coupling reaction.

### 3.2. The Thermo-Sensitivity and Optical Properties of Poly-N-CQDs

The excitation and fluorescence emission spectra demonstrate that the poly_14_-N-CQDs displays maximum excitation and emission peaks at 380 nm and 451 nm, respectively. Figure 3a–d present the fluorescence emission spectra of the poly_14_-N-CQDs with a concentration of 0.5 wt% at 340 nm, 360 nm, 380 nm and 400 nm excitation wavelength from 15 to 50 °C. The emission peaks locate at 451 nm when the excitation wavelengths increase from 340 to 380 nm at different temperatures, indicating an excitation- and temperature-independent PL behavior, although the emission wavelength slightly red-shifts when the excitation wavelength is at 400 nm. As the excitation wavelength increases, the fluorescence emission intensity of the poly_14_-N-CQDs first increases, followed by a decrease, and reaches the maximum emission of the poly_14_-N-CQDs at an excitation wavelength of 380 nm.

PNIPAAm is well-known as a typical thermo-responsive polymer, which displays a coil-globule transition at a lower critical solution temperature of around 32 °C [25,26]. Figure 3c clearly displays that fluorescence intensity gradually becomes weaker from a highly transparent solution (40 °C, below LCST) to an opaque solution (50 °C, above LCST) when the temperature increases from 15 °C (below LCST) to 50 °C (above LCST). At lower temperature, the PNIPAAm chains become highly hydrated and exhibit a relative stretching conformation due to the predominately intermolecular hydrogen bonding between the PNIPAAm chains and the surrounding water molecules in the dilute aqueous solution, and the N-CQDs are uniformly dispersed and fully exposed in the homogeneous solution, which exhibits a maximum PL intensity at 15 °C. With the increase in temperature, the stretched PNIPAAm chains appear shrunken into a coiling configuration due to the break in intermolecular hydrogen bonding with water molecules and the formation of intramolecular hydrogen bonding between the -NH- and C=O groups in the PNIPAAm chains, resulting in a weaker PL intensity of the N-CQDs. When the temperature is elevated above LCST, the intramolecular hydrogen bonding between the -NH- and C=O groups results in a compact and collapsed conformation of PNIPAAm chains, the solution becomes opaque and heterogeneous, and the N-CQDs are embedded in aggregated PNIPAAm chains, resulting in an obvious decrease in PL intensity.

Figure 3e shows the fluorescence emission spectra of 0.5 wt% aqueous solution of poly_7_-N-CQDs at a 380-nm excitation wavelength and different temperatures. As compared, Appendix A present the fluorescence emission spectra of the poly_14_ and the mixture of the poly_14_ and N-CQDs at the same conditions. As seen from Figure 3a–e and Appendix A, a weak scattering peak exists at the excitation wavelength whenever the excitation wavelength changes, which is ascribed to Rayleigh scattering [27]. Below LCST, the dilute aqueous solution of poly_14_, the mixtures of poly_14_ and N-CQDs, poly_7_-N-CQDs and poly_14_-N-CQDs are all transparent, the PNIPAAm chains become hydrated and display a stretching or coiling conformation, and the particle size of the N-CQDs is only in the range from 4 to 7 nm, which is far less than 1/10 of the excitation wavelength. With the increase in the temperature, the samples display different coil–globule transition behaviors. The poly_14_ and the mixture of poly_14_ and N-CQDs suddenly changes from transparent solutions into opaque solutions when the temperature increases from 30 °C (below LCST) to 32 °C (above LCST); the poly_7_-N-CQDs also transforms from a transparent solution (35 °C, below LCST) into an opaque solution (39 °C, above LCST) when the temperature increases, and the poly_14_-N-CQDs turns from a transparent solution (40 °C, below LCST) into an opaque solution (50 °C, above LCST) at the highest temperatures. It is interesting to notice that the above-mentioned abrupt transitions of the solutions from transparent to opaque coincide with the dramatic changes in the scattering peak intensity in response to the temperature changes, implying that the sharp increase in the scattering peak intensity results from the phase transition of the PNIPAAm chains in different samples. When the temperature is above LCST, PNIPAAm chains become dehydrated and exhibit a shrunken globule conformation [28], the solution becomes opaque and heterogeneous. The particle sizes of the dilute aqueous solutions of different samples (above LCST) were measured, as depicted in Figure 3g. The globule particle sizes range from 70 to 440 nm, 90 to 520 nm and 100 to 700 nm for poly_14_, poly_7_-N-CQDs and poly_14_-N-CQDs, respectively. The particle sizes are close to the excitation wavelengths, indicating a Mie scattering pattern [27,29]. The intensity of the scattering peaks rises with the increase in globule sizes resulting from the elevating temperature. Figure 3f shows the plot of the maximum scattering peak intensity of different samples as a function of temperature. It is clearly observed that the poly_14_-N-CQDs displays the same phase transition behaviors at different excitation wavelengths. The phase transition temperatures of the poly_14_, poly_7_-N-CQDs and poly_14_-N-CQDs are around 32 °C, 37 °C and 47 °C, respectively. To confirm the phase transition behaviors of the above-mentioned samples, the transmittance of poly_14_, poly_7_-N-CQDs and poly_14_-N-CQDs with a concentration of 0.5 wt% was measured at different temperatures, as shown in Appendix A. The cloud points of the poly_14_, poly_7_-N-CQDs and poly_14_-N-CQDs are 31 °C, 36 °C and 47 °C, respectively. The results obtained from the fluorescence emission spectra are in conformity with cloud points measured by UV-Vis spectro-photometer. The introduction of N-CQDs as pendent groups onto the PNIPAAm chains can regulate the phase transition temperature because the N-CQDs as pendant groups suppress the coil–globule transition of the PNIPAAm chains.

Furthermore, it is demonstrated that the changes in the scattering peak intensity and the emission peak intensity response to temperature were reversible, as shown in Figure 4. When the temperature was below LCST, the scattering peaks and the emission peaks shifted rapidly back to their original positions from their positions at T > LCST. The excellent reproducibility of the responses from poly-N-CQDs to temperature makes it possible to develop an N-CQDs-based thermosensitive device or sensor. The mechanism of the PL intensity to the thermal responsiveness is proposed as depicted in Figure 2.

The fluorescence emission spectra of 0.1 wt% and 1 wt% of the poly_14_-N-CQDs solutions are also investigated, as shown in Appendix A, and the plot of the maximum scattering peak intensity as a function of the temperature is presented in Figure 3f. The phase transition temperatures are about 52 °C, 47 °C and 43 °C when the poly_14_-N-CQDs concentrations are 0.1 wt%, 0.5 wt% and 1 wt%, respectively. With an increasing concentration of poly_14_-N-CQDs, its phase transition temperature decreases, and the FL emission intensity increases.

### 3.3. The Dependency of the Fluorescence Intensity on the pH

The stability of the poly_14_-N-CQDs in the different pH conditions is also investigated at 25 °C (as shown in Figure 5a. It is noteworthy that the as-prepared poly_14_-N-CQDs shows strong stability in fluorescence activity with no damage in a wide range of pH values from 3 to 10; even in a strongly acidic environment (pH 1–2), there is still enough fluorescence intensity. The thermo-sensitivity and optical properties of the poly_14_-N-CQDs with a 0.5 wt% aqueous solution at pH 2.0, pH 7.0 and pH 12.0 and different temperatures are studied, as shown in Appendix A. Figure 5b exhibits the phase transition behaviors of the poly_14_-N-CQDs solutions at pH 2.0, pH 7.0 and pH 12.0 and different temperatures. As seen in Figure 5b, the phase transition temperatures at pH 2.0 and pH 7.0 are about 30 °C and 37 °C, respectively. The phase transition temperature at pH 2.0 is even lower than that of pure poly_14_, and the phase transition temperature at pH 7.0 is the same as that of aqueous solution. However, no obvious phase transition occurs even when the temperature reached 70 °C at pH 12.0, and the maximum emission peak also displays an obvious red-shift. The phase transition behaviors and PL intensity variations are ascribed to the protonation and deprotonation of the fluorescent functional groups on the surface of the N-CQDs [30]. Zhou et al. [31] also reported the same trend of an increase in the PL intensity of neat CQDs with a rise in the pH from 1.5 to 7. At a lower pH value, the carboxylic groups on the surface of CQDs are protonated, and hydrogen bonding interactions between the -COOH and -CO-NH- groups from the PNIPAAm chains might be conducive to the coiling–globule transition at a lower temperature, resulting in a decrease in the phase transition temperature [32]. With an increase in pH, the -COOH groups are deprotonated, whereby a “protective shell” with a negative charge is gradually formed on the surface of the N-CQDs with a lower non-radiative recombination rate [33]. Furthermore, the electrostatic repulsion force restricts the coiling–globule transition, even leading to the disappearance of the phase transition at a higher pH (pH 12.0). These results demonstrate that the pH of the solution can not only affect the surface state of N-CQDs and alter the band edge resulting in different optical properties [34], but also influence the coil–globule transition of PNIPAAm chains, which adjusts the phase transition temperature of PNIPAAm chains.

### 3.4. The Poly-N-CQDs in Different Solvents for Solvent Recognition

Although as-fabricated N-CQDs are easily dissolved in water, it is difficult to disperse in some organic solvents. It is observed that the poly-N-CQDs could also readily dissolve in many organic media, which broadens the practical applications of N-CQDs in organic media. The fluorescent properties of the poly-N-CQDs in different solvents are studied. Figure 6a shows the fluorescence spectra of the poly_14_-N-CQDs in N, N-Dimethylformamide (DMF), THF, ethanol, acetone and water using 380 nm as the excitation wavelength. The fluorescence emission peaks and the fluorescence intensity can significantly change in different solvents. The emission peaks of the poly-N-CQDs in DMF, THF, ethanol, acetone and water are 435 nm, 434 nm, 443 nm, 437 nm and 452 nm, respectively. The emission peaks of the poly-N-CQDs in DMF, THF and acetone are similar, but they very different from that of the poly_14_-N-CQDs in ethanol and water. The fluorescence intensity is strongest in the THF, and then in the water, DMF, ethanol and acetone in sequences. Several reasons can be considered: The first is that the solubility of the poly_14_-N-CQDs in various solvents is different. The poly_14_-N-CQDs dissolves easily in THF, DMF and water, and the polymer chains of the poly_14_-N-CQDs are fully extended. The solubility of the poly_14_-N-CQDs becomes relatively weaker in the ethanol and acetone, and the polymer chains become coiling and winding, resulting in a decrease in fluorescence intensity. The second is that the -COOH, -NH_2_ and -OH functional groups exist on the surface of the N-CQDs, which could form the hydrogen bonding interactions with the -CO-NH- groups of the polymer chains; the energy transfers between the N-CQDs and the polymer may lead to fluorescence intensity changing [20]. The last reason is that the different surface structure of N-CQDs imposed by different medium environments also causes their various fluorescent properties [34]. The above-mentioned effects bring about variations in fluorescent properties. Various emission colors in different solvents are visible to the naked eye under a 365-nm UV light (Figure 6c). The corresponding CIE coordinates (Figure 6d) are (0.1585, 0.0935), (0.1558, 0.0808), (0.1568, 0.0811), (0.155, 0.1034) and (0.1526, 0.1324) for the THF, DMF, acetone, ethanol and water, respectively.

### 3.5. The Poly-N-CQDs as Fluorescence Probe for Fe^3+^ Ion Detection

N-CQDs have been vastly investigated and studied in the environmental and biomedical fields due to their strong fluorescent properties and high photostability. Herein, the poly-N-CQDs is applied as a fluorescence probe in the detection of Fe^3+^ ions. For this experiment, the sensitivity of the poly_14_-N-CQDs to various metal ion salts was investigated, as shown in Appendix A. A change in the PL intensity (F/F_0_) of the poly_14_-N-CQDs at 451 nm in the presence of different metal ions is exhibited in Figure 7a, where F and F_0_ are the PL intensities at 451 nm in the presence and absence of metal ions, respectively. A significant fluorescent quenching is observed when the concentration of Fe^3+^ is 4 mM, while the same concentration of other metal ions (Na^+^, Mg^2+^, K^+^, Ca^2+^ and Al^3+^) displays no obvious changes. This result exhibits high selectivity of the as-prepared poly_14_-N-CQDs toward Fe^3+^ as compared with other metal ions. To evaluate the sensitivity, different concentrations of Fe^3+^ in the range of 0.025–4 mM are investigated. Figure 7b shows that the fluorescence intensity at 451 nm decreases as the Fe^3+^ concentration gradually increases, indicating the ability of poly_14_-N-CQDs to sense a change in the Fe^3+^ concentration. The fluorescence quenching data follows the Stern–Volmer equation [35]. The Stern–Volmer plot in Figure 7c shows that an adequate linear fit with a correlation coefficient (R^2^) of 0.997 between the PL intensity and the Fe^3+^ concentrations in a wider range of 0.025–1 mM, as compared to the other reported literature [36,37,38]. The limit of detection (LOD) of 9.49 μM is calculated according to the International Union of Pure and Applied Chemistry 3σ criterion (DL = 3SD/S, where SD is the standard error of the intercept, and S is the slope of the calibration curve); it is slightly higher than the reported values of native CQDs in the literature [36,37,38], implying that the possible mechanism of fluorescence quenching may be attributed to the aggregation of the N-CQDs, since the PNIPAAm chains could suppress the aggregation of the N-CQDs due to the structural nature of the poly-N-CQDs. Nevertheless, as aforementioned, poly-N-CQDs simultaneously exhibits a tunable phase transition temperature in PNIPAAm components and luminescent properties in N-CQDs. These thermosensitive and luminescent hybrid nanomaterials have potential applications as fluorescence sensors for the detection for ions or biomolecules; more importantly, they could also be applied in extraction and purification of ions or biomolecules.

## 4. Conclusions

Poly-N-CQDs with thermo-sensitivity and luminescent properties were successfully synthesized. The phase transition behaviors and PL intensity can be modulated by altering the composition and concentration of the poly-N-CQDs as well as the temperature and pH of the aqueous medium. The different fluorescent properties of the poly-N-CQDs in THF, DMF, acetone, ethanol and water can be employed as a fluorescence sensor for solvent recognition. The selectivity and sensitivity of the poly-N-CQDs to Fe^3+^ ions can be used as a fluorescence sensor for Fe^3+^ detection. These multifunctional hybrid polymer nanomaterials have potential applications as an N-CQDs-based thermo-sensitive device or sensor.

## Data Availability

Not applicable.

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
