# Peer review of "Stimulus-Responsiveness of Thermo-Sensitive Polymer Hybridized with N-Doped Carbon Quantum Dots and Its Applications in Solvent Recognition and Fe^3+^ Ion Detection"

_polymers, 2022, doi:10.3390/polym14101970_

Round 1

Reviewer 1 Report

The manuscript describes the synthesis and characterization of hybrid fluorescent polymers containing N-doped carbon quantum dots.  The synthesis and fluorescence of the hybrid polymers are well described and various applications of these polymers are suggested such as fluorescence-based sensor for Fe3+. Some specific comments and suggestions are given below:

  1. The title for section 3.2 would be clearer as “The thermo-sensitivity and optical properties of poly-N-CQDs”. Please, remove “at different temperatures”.
  2. 3 The dependency of the fluorescence intensity on the pH
  3. When discussing the fluorescent properties in different solvents, please, change the “dissolving capacity” of the solvent to the “solubility of the polymer in various solvents”. If the solubility in some solvents is lower, the intensity of fluorescence may be lower but these solvents may not be used for solvent recognition. The formation of hydrogen bonds may be a better reason.
  4. Maybe the size distribution should be moved to the main text.
  5. The graphs in the supplementary cannot be seen!

Reviewer 2 Report

Chen and Zhao report on the synthesis of a hybrid copolymer containing a poly(NIPAAm-co-AAc) random copolymer backbone and fluorescent carbon dots covalently attached as side groups. The authors studied the thermoresponsive properties of the hybrid material, its photophysical response to temperature changes and provided proof of its utilization as sensor material for selective Fe3+ ion detection. The work is original and interesting. However, the authors should take into account the comments below.

  1. The yield of the coupling reaction should be reported.
  2. It would be interesting to study copolymers with different degrees of carbon dot functionalization. Changes in the thermoresponsive and fluorescence properties may be observed and regulated accordingly.
  3. Figure 3: Images of lower magnification/resolution showing a larger number of particles should be presented.
  4. Fig. 6: no stretched polymer conformations are expected at low temperatures. Below LCST chains should be in a coil conformation. Please discuss.
  5. Use of English should be improved in several parts of the manuscript.

Author Response

请看附件
